# Preliminary Results of NGS Gene Panel Test Using NSCLC Sputum Cytology and Therapeutic Effect Using Corresponding Molecular-Targeted Drugs

**DOI:** 10.3390/genes13050812

**Published:** 2022-05-02

**Authors:** Kei Morikawa, Kohei Kinoshita, Hirotaka Kida, Takeo Inoue, Masamichi Mineshita

**Affiliations:** Department of Internal Medicine, Division of Respiratory Diseases, St. Marianna University School of Medicine, Kawasaki 216-8511, Japan; kohei.kinoshita@marianna-u.ac.jp (K.K.); h2kida@marianna-u.ac.jp (H.K.); t2inoue@marianna-u.ac.jp (T.I.); m-mine@marianna-u.ac.jp (M.M.)

**Keywords:** epidermal growth factor receptor mutation, KRAS mutation, lung cancer compact panel, MET exon14 skipping, next generation sequencing, non-small cell lung cancer, sputum cytology

## Abstract

As more molecular-targeted drugs for advanced non-small cell lung cancer are brought to market, batch tests for the identification of gene mutations are needed at initial diagnosis. However, since current gene panel tests require a sufficient amount of tissue sample, there are many instances where panel tests cannot be performed. Therefore, we have developed a highly sensitive next generation sequencing (NGS) panel test to facilitate cytological specimens. Herein, we describe three cases positive for epidermal growth factor receptor (EGFR) exon 19 deletion, MET exon 14 skipping, and KRAS G12A using NGS analysis from sputum. In each case, genetic information was consistent with companion diagnostic analysis obtained from tissue samples collected under bronchoscopy. In cases of EGFR and MET mutations, the corresponding tyrosine kinase inhibitors were highly effective. This is the first report to demonstrate that a novel panel test could detect gene mutations in sputum samples in clinical practice and compare the gene allele ratio with the sample directly collected from the lesion.

## 1. Introduction

Gene mutation detection has become increasingly important for non-small cell lung cancer (NSCLC) treatment strategies [1]. On the other hand, as gene mutation panel tests need high-quality tissue samples, it can be challenging to perform batch testing when sample collection is difficult or if only a small amount of tissue is available. We report promising preliminary results on the development of a lung cancer compact panel (LC-CP) for cytological specimens and its application for sputum cytology (HREC ID 4814).

## 2. Methods

A prospective observational study of gene panel analysis using cytological specimens is currently underway for Japanese patients with suspected lung malignancies. Cytological specimens such as transbronchial biopsy (TBB) brushing, transbronchial needle aspiration (TBNA), and pleural effusion were used for genetic panel testing. This study had registered 260 specimens from May 2020 to November 2021 (HREC ID 4814). Among these, sputum was collected for the gene panel prior to bronchoscopy in 3 patients who were found to be positive for sputum cytology.

Sputum and cytological specimens were collected in a sample container (GM tube, Genemetrics, Osaka, Japan) which contains a nucleic acid stabilizer, and were processed with the Maxwell^®^ RSC Blood DNA Kit and Maxwell^®^ RSC simply RNA Cells Kit (Promega, WI, USA). Using the purified nucleic acid, a lung cancer compact panel (DNA Chip Research Inc., Tokyo, Japan) NGS assay was performed. The lung cancer compact panel (LC-CP) is an amplicon-based high-sensitivity NGS panel capable of measuring eight druggable genes (EGFR, BRAF, KRAS, ERBB2, ALK, ROS1, MET, RET) for lung cancer. The experimental process is briefly described below. As a starting material, 5 ng of DNA (double stranded DNA) and 5 ng of RNA were used for the assay of each module. The LC-CP is composed of two DNA modules and two RNA modules. Therefore, 10 ng of dsDNA and 10 ng of RNA were set as minimum requirement of starting input material (i.e., the yield of purified nucleotides). For the DNA assay, multiplex PCR using KOD-Plus-Neo (Toyobo, Osaka, Japan) was performed to amplify EGFR (exon 18–21), BRAF (exon 15), KRAS (exon 2), ERBB2 (exon 8, 17, 20), and MET (near exon 14). Two DNA panels (DNA module 1 and DNA module 2), were designed and optimized to detect somatic mutation sensitively and quantitatively with unbiased amplification of these hotspot regions. Forty cycles of 98 °C for 10 s and 62 °C for 30 s were performed to amplify regions on the panel of the DNA module. For RNA assays, first strand cDNA was synthesized by Revertra-ace^®^ (Toyobo), and multiplex PCR using KOD Fx Neo (Toyobo) was performed to detect ALK, ROS1, RET fusion gene variants, and MET exon 14 skipping. Before cDNA synthesis, input RNA was mixed with 9-base random primer (Toyobo) and incubated at 65 °C for 5 min for denaturing RNA and for hybridization with primer. The reaction mixture was incubated at 30 °C for 10 min, and then at 42 °C for 60 min in the cDNA synthesis step. Forty cycles of 98 °C for 15 s, 60 °C for 30 s, and 68 °C for 10 s, followed by extension at 68 °C for 1 min, were performed to amplify target regions. After purification with AMPure XP (Beckman Coulter Life Sciences, Indianapolis, IN, USA), sequence libraries from these PCR products were prepared using the GenNext^®^ NGS Library Prep Kit (Toyobo). All steps were performed according to the manufacturer’s protocols. The constructed sequence library was sequenced using MiSeq (Illumina, CA, USA) by paired-end mode (2 × 150 bp). Bioinformatics analysis pipeline is described in the flowchart of Figure 1. The Illumina adapter sequences were trimmed by Trimmomatic v 0.33 and paired-end sequences were joined by FLASH v 1.2.11 fastq joining tool. The joined reads were mapped on the target regions of the human genome by BWA aligner v 0.7.17 and mutation variant was called by analyzing bam format alignment output by custom programming scripts. Reference sequence for variant call was constructed based on the hg19 human reference genome. HGVD polymorphism database information was used for the discrimination between SNP polymorphism and somatic mutation of variant of unknown significance (VUS).

## 3. Case Reports

Sequencing depths and quality control metrics data of NGS results are summarized in Appendix A. 

Case 1. A male in his 60 s with a smoking history (Brinkman index of 850) was referred to our hospital complaining of cough and blood in sputum over several months. A diagnosis of class V adenocarcinoma was made by sputum cytology. A second sputum sample was collected and bronchoscopy was performed. The left main bronchus and basal trunk showed cancerous lymphangiopathy on endoscopy. The brushing cytology sample collected from the left B^6^ of the primary lesion was submitted for LC-CP, and the biopsy tissue underwent Oncomine™ Dx Target Test. A diagnosis of adenocarcinoma, cT3N3M1c (BRA), stage IVB was revealed (Figure 2). Six days after bronchoscopy, LC-CP confirmed EGFR exon19 deletion by brushing cytology sample (DNA integrity number (DIN) value 8.0, DNA yield 320.8 ng, Allele frequency (AF) 35.5%). Similarly, exon19 deletion was detected by LC-CP from the collected sputum sample (DIN value 6.8, DNA yield 1336 ng, AF 7.4%) (Figure 3). Oncomine™ Dx Target Test confirmed these results (EGFR p. Glu746_Ala750del (COSM ID 6223) with a mutation frequency of 28.7%) using tissue specimen. Osimertinib, an EGFR- tyrosine kinase inhibitor (TKI), was administered with remarkable results (Figure 4).

The analytical performance of LC-CP was thoroughly validated according to the ICH guideline (https://www.pmda.go.jp/files/000156867.pdf accessed on 1 April 2022). The LC-CP is currently under review by pharmaceutical affairs (Pharmaceuticals and Medical Devices Agency, Tokyo, Japan) and is expected to be approved for clinical practice in Japan as a companion diagnostic kit soon. Analytical performance, clinical validity and details of the analysis of the LC-CP can also be found in this reference [2].

Case 2. A male in his 80 s with a smoking history (Brinkman index of 1000) was referred to our hospital complaining of blood in sputum. A diagnosis of class Ⅳ adenocarcinoma was made by sputum cytology. A second sputum sample was collected and bronchoscopy was performed. During bronchoscopic examination, bloody sputum was observed at the upper right lobe where the primary lesion was present. The brushing cytology sample collected from the right B^1^a was submitted for LC-CP, and the biopsy tissue underwent Archer^®^ MET Dx Target Test. A diagnosis of adenocarcinoma, cT4N2M0, stage ⅢB was determined (Figure 5). Further, a diagnosis of MET exon14 skipping was detected by LC-CP from the brushing cytology sample (DIN value 8.8, DNA yield 532 ng, AF 39.7%). This same mutation was detected by LC-CP from the collected sputum sample (DIN value 8.2, DNA yield 3160 ng, AF 0.7%). Archer^®^MET Dx Target Test confirmed these results using tissue specimen. Tepotinib, a MET-TKI, was administered and showed immediate therapeutic effects (Figure 6).

Case 3. A male in his 80 s with a smoking history (Brinkman index of 2500), was unintentionally found to have lung cancer by CT scan performed before surgery for an inguinal hernia. A diagnosis of class Ⅴ adenocarcinoma was made by sputum cytology. A second sputum sample was collected and bronchoscopy was performed. During bronchoscopic examination, bloody sputum was observed at the right lower lobe where the primary lesion was present. The brushing cytology sample collected from the right B^9^a was submitted for LC-CP, and a diagnosis of adenocarcinoma, cT4N1M0, stage ⅢA was determined (Figure 7). Further, a diagnosis of KRAS G12A was detected by LC-CP from the brushing cytology sample (DIN value 9.5, DNA yield 3680 ng, AF 81.9%). This same mutation was detected by LC-CP from the collected sputum sample (DIN value 8.2, DNA yield 684 ng, AF 9.9%).

## 4. Discussion

Sputum cytology is often useful to detect early lung cancer endoscopically, such as bronchial carcinoma in situ, when image detection is difficult. It has a sensitivity of 66% and a specificity of 99%, and is a pathologically established diagnostic method [3]. It has been reported that squamous cell carcinoma, which is closely related to a history of heavy smoking, is frequently detected. However, there are a certain number of cases in which sputum cytology is positive, such as small cell lung cancer and adenocarcinoma with cancerous lymphangiopathy, regardless of histological type [4,5]. From a cancer stage perspective, the sensitivity of sputum cytopathology is higher in advanced cancer than in early-stage cancer [6]. Pathologically, spread through air spaces (STAS) is considered to be a positive factor for sputum cytopathology [7]. Clinically predictive factors for STAS include malignant lymphangiopathy, tumor diameter > 10 mm, and high standardized uptake value (SUV) for 18F-fluorodeoxyglucose (FDG)-positron emission tomography (PET) image [8]. In this study, we described three cases of different clinicopathological conditions that tested positive for sputum cytopathology. Case 1 showed cancerous lymphangiopathy on endoscopic findings, and case 2 displayed bloodied sputum from the primary lesion without endoscopic lymphangiopathy, while case 3 revealed weak uptake of FDG-PET and CT images suggesting a mucous-producing tumor with glass nodule. There are cases where sputum cytopathology is positive even when it does not correspond to clinical predictors of STAS. Furthermore, we found it significant that it was not only EGFR mutations detected, but also rarer mutations such as MET and KRAS were found in sputum specimens.

Related research concerning sputum cytology is broadly divided into the development of equipment for improving the detection sensitivity of endoscopic early-stage lung cancers [9] and the highly sensitive nucleic acid detection using sputum, often related to microRNA [10]. However, the latter is usually detected by PCR method. Recently, gene mutation detection using sputum has been reported, but these were also detected by the digital droplet PCR method [11,12]. Although there have been reports of NGS analysis of sputum using ctDNA [13,14], our report is the first to include cases where gene panel analysis was performed on both DNA and RNA, and where the therapeutic effects of molecular-targeted drugs were observed.

Our first case was accompanied by bloodied sputum on the first visit, and CT image revealed cancerous lymphangiopathy. From the smear sample obtained by sputum cytology, about 10 agglomerates composed of 20 to 70 malignant cells appeared (Figure 2), while other inflammatory cells such as squamous epithelial cells, dust cells, and neutrophils were present. Both DNA integrity number (DIN) and RNA integrity number (RIN) values were high without any additional sputum collection or storage procedures. Similarly, for case 2 and 3, a small amount of gene mutation was detected from sputum. For all cases, sputum specimens tended to show lower gene allele frequency when compared to brush specimens when cells were collected directly from the lesion under bronchoscopy. Therefore, it is considered that high sensitivity is an essential requirement for NGS analysis in sputum samples [15].

In this report, a major limitation was that gene panel tests for sputum were only performed in cases where sputum cytology was already confirmed positive. The reason for this was that the sputum examination had already been performed at the referral hospital. After sputum collection, malignant cells were collected from the primary lesion by bronchoscope, and the mutant genes were matched in all cases. However, the AF of gene mutations remained low compared to the samples collected by bronchoscopy, which might be considered unfavorable from the viewpoint of sensitivity. It has been reported that tumor cells account for less than 1% of the cells in sputum [15]. Moreover, since sputum contains mostly normal cells, the technical aspect of a cytopathologist might be affected [16]. Therefore, standardization for the handling of sputum samples may be required in the future.

Currently, mainstream gene panel tests use tissue specimens and blood tests, but if the sputum cytopathology is positive, there are instances when high-sensitivity NGS analysis using sputum specimens is possible, as the least invasive ways for the patients.

## 5. Conclusions

Our new panel test was able to detect gene mutations from sputum samples in clinical practice and we could compare the gene allele ratio with the sample directly collected from the lesion. In the future, a large-scale study testing sputum samples before bronchoscopy will be necessary to confirm diagnosing accuracy.

## Figures and Tables

**Figure 1 genes-13-00812-f001:**
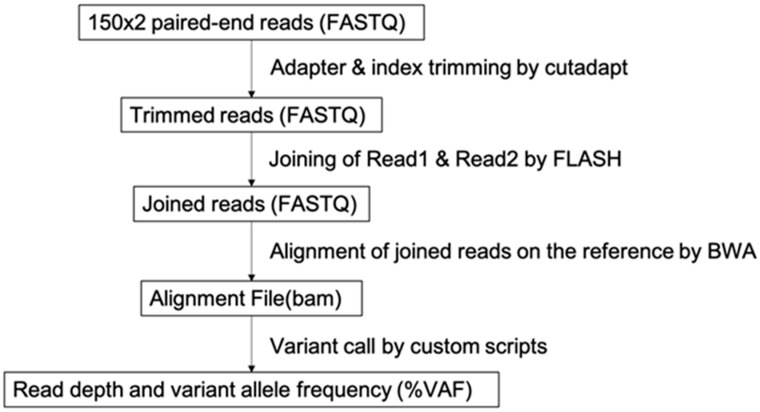
Flowchart of bioinformatics pipeline.

**Figure 2 genes-13-00812-f002:**
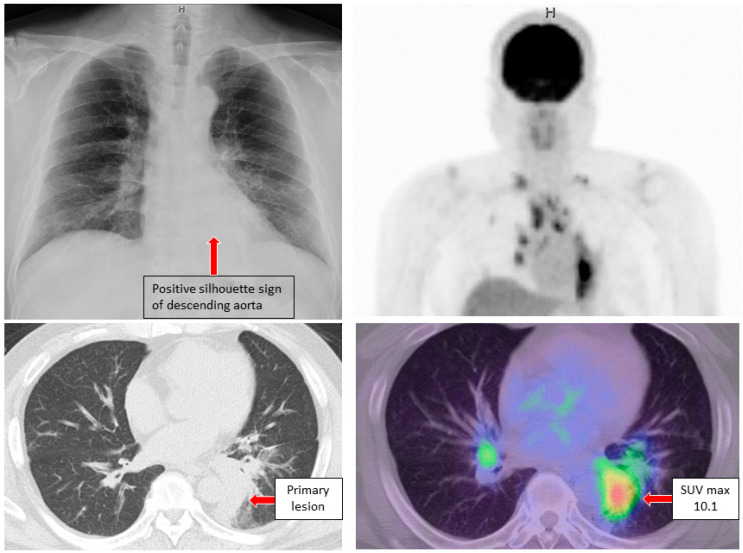
Chest X-ray, CT and FDG-PET examinations in case 1 revealed primary lung cancer at the left lower lobe (red arrow) and multiple lymph node metastases. CT: computed tomography, FDG-PET: 18F-2-fluoro-2-deoxy-D-glucose-positron emission tomography.

**Figure 3 genes-13-00812-f003:**
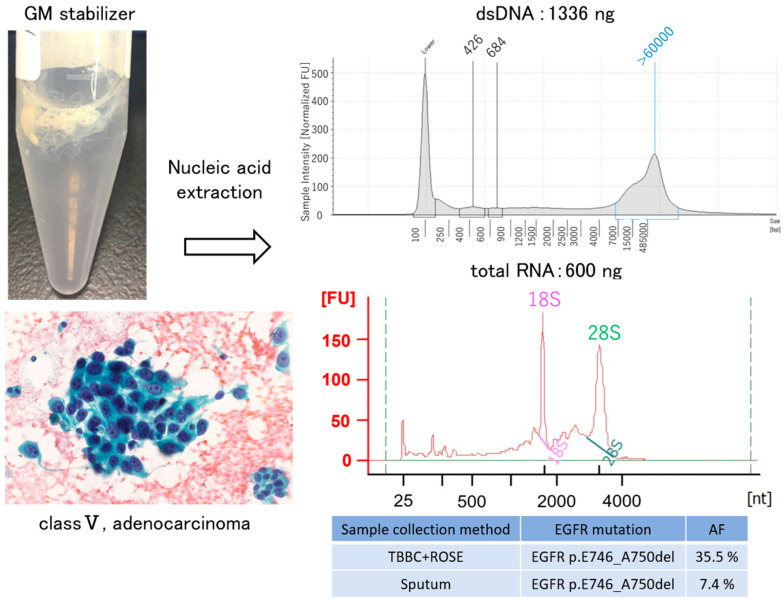
Sputum containing malignant cells was injected into the GM tube, and the nucleic acid was extracted. Sufficient nucleic acid yields were obtained for both DNA and RNA. EGFR activating mutations were also detected, although the AF was lower than bronchoscopic brush samples (case 1). TBBC: transbronchial brushing cytology. ROSE: rapid on-site cytologic evaluation. AF: allele frequency.

**Figure 4 genes-13-00812-f004:**
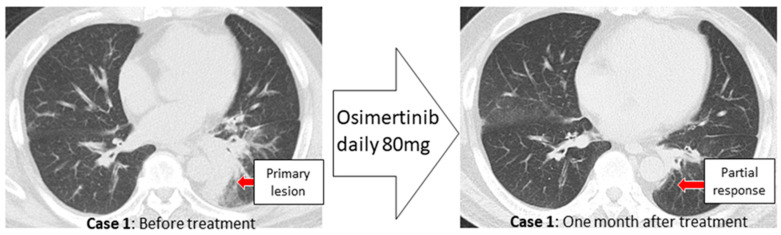
Cases 1 showed a favorable response after administration of molecular-targeted drugs.

**Figure 5 genes-13-00812-f005:**
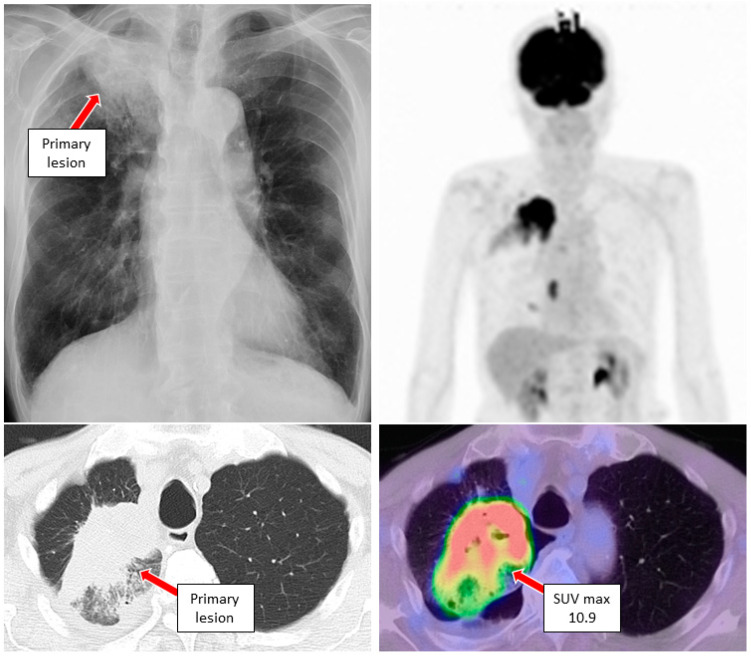
Chest X-ray, CT and FDG-PET examinations in case 2 revealed primary lung cancer at the right upper lobe. CT: computed tomography, FDG-PET: 18F-2-fluoro-2-deoxy-D-glucose-positron emission tomography.

**Figure 6 genes-13-00812-f006:**
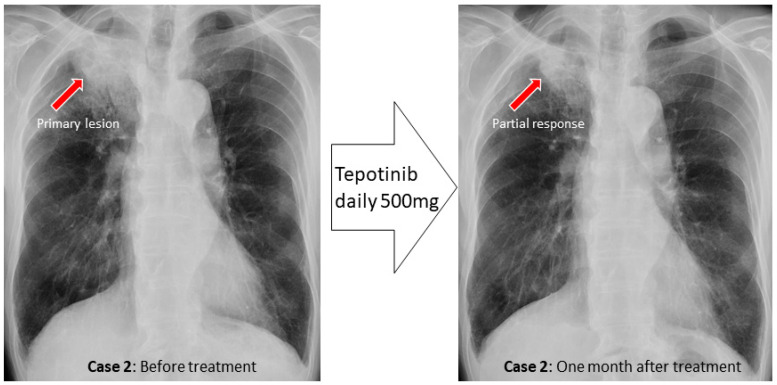
Cases 1 and 2 showed a favorable response after administration of molecular-targeted drugs.

**Figure 7 genes-13-00812-f007:**
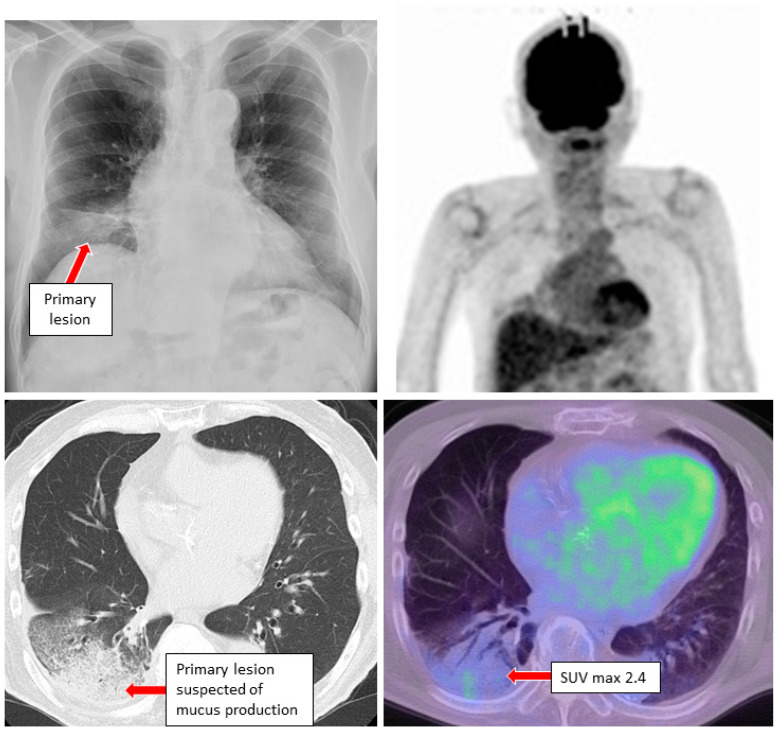
Chest X-ray, CT and FDG-PET examinations in case 3 revealed primary lung cancer at the right lower lobe. CT: computed tomography, FDG-PET: 18F-2-fluoro-2-deoxy-D-glucose-positron emission tomography.

## Data Availability

DDBJ Sequence Read Archive (DRA), [Submission ID] keimorikawa-0001, [Hold date] 30 December 2022; [Accession number] Submission: DRA013115 (keimorikawa-0001_Submission) BioProject: PRJDB12609 (PSUB016194) BioSample: SAMD00424683-SAMD00424686 (SSUB019900) Experiment: DRX320506-DRX320509 (keimorikawa-0001_Experiment_0001-0004) Run: DRR331502-DRR331505 (keimorikawa-0001_Run_0001-0004).

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
