# Peer review of "Preliminary Results of NGS Gene Panel Test Using NSCLC Sputum Cytology and Therapeutic Effect Using Corresponding Molecular-Targeted Drugs"

_genes, 2022, doi:10.3390/genes13050812_

Round 1

Reviewer 1 Report

The authors provided a manuscript on the evaluation of gene mutation using next genome sequencing in sputum samples to identify mutations in lung cancer.

I agree that data may be very interesting, in particular if the clinical prospective trail may confirm preliminary results exposed in this manuscript.

However, there some issues that need to be addressed to increase the value of the manuscript.

1- figure 5 must be showed as the last of those presented. I did not understand why they preferred to put in the middle of them. Please change numbers of position

2- in figures I suggest to indicate with a symbol mark the tumor mass to be more specific in their explanation

3- in figure 4 it appear that heart uptake FDG more than lung tumor. Please add a more clear picture and use symbol mark to identify tumor

4- why the third case was not treated for lung cancer? KRAS G12A mutation has a poor prognosis but a treatment is also useful. Please comment more on outcome of this case

Author Response

Answer to reviewer 1

The authors provided a manuscript on the evaluation of gene mutation using next genome sequencing in sputum samples to identify mutations in lung cancer.

I agree that data may be very interesting, in particular if the clinical prospective trail may confirm preliminary results exposed in this manuscript.

However, there some issues that need to be addressed to increase the value of the manuscript.

Thank you for pointing out many details that were missing in this paper. I will answer the points from the reviewer one by one.

1- figure 5 must be showed as the last of those presented. I did not understand why they preferred to put in the middle of them. Please change numbers of position

Answer: We correct the image position.

2- in figures I suggest to indicate with a symbol mark the tumor mass to be more specific in their explanation

3- in figure 4 it appear that heart uptake FDG more than lung tumor. Please add a more clear picture and use symbol mark to identify tumor

Answer: We added the explanation in the figure. Since case 3 was a mucinous producing tumor and the accumulation of FDG-PET was really low. Hence, we added SUV max description.

4- why the third case was not treated for lung cancer? KRAS G12A mutation has a poor prognosis but a treatment is also useful. Please comment more on outcome of this case

Answer: Other than KRAS G12C, molecular-targeted drugs could not be administered because that are not covered by medical insurance in Japan. Of course, we did anti-cancer drug treatment, but since it was not a case that led to a molecular-targeted drug. Therefore, it was not possible to post the image after administration of molecular-targeted drugs.

Reviewer 2 Report

The authors should consider the followings:

  1. In methodology, the authors should elaborate the workflow of the NGS assay performed, and the bioinformatic pipelines used, preferably presented as a workflow diagram(s).
  2. The authors should provide the respective sequencing depths and QC values of the NGS assays used.
  3. Did the cases gene sequence evaluate with the respective ethnicity of the subject?
  4. Did the NGS panel follow the ACMG guidance, or other authorities? If yes, which guidance did the panel follow? Please add information of the guidance followed.
  5. Please provide information of the ethnicity of the subjects, if available.
  6. The authors should give details (in each case) the turnaround time of initial diagnosis to result reporting of the NGS test.
  7. In case 1, please provide the Oncomine Dx Target Test results by the tissue specimen.
  8. The authors may give the information of the PPV and NPV (positive and negative predictive values) of the assay in this study, and further comments of them.
  9. The authors should add the area of limitation, using sputum cytology, and please list the limitation of the current study as a whole.
  10. The authors may improve the figure resolution of Figure 2.
  11. In Figure 5, please indicated the time of capture after treatment.
  12. The authors may increase the number of relevant references in the article.
  13. As for the case presentation of case 1 to 3, the authors may use a flowchart and timeline to better illustrate the treatments (types, doses and frequency).
  14. The authors should clearly mention the novel findings of the research, in the part of abstract and conclusion.

Author Response

The Authors would like to thank the Reviewer for raising many details that were missing in our paper. Point-by-point answers are given to all questions and comments below.

The authors should consider the followings:

  1. In methodology, the authors should elaborate the workflow of the NGS assay performed, and the bioinformatic pipelines used, preferably presented as a workflow diagram(s).

A1) We added a more detailed description of the NGS assay in Methods section, (page 5).

“The experimental process is briefly described below. As a starting material, 5ng of DNA (double stranded DNA) and 5ng of RNA were used for the assay of each module. The LC-CP is composed of two DNA modules and two RNA modules. Therefore, 10ng of dsDNA and 10ng of RNA were set as minimum requirement of starting input material (i.e., the yield of purified nucleotides). For the DNA assay, multiplex PCR using KOD-Plus-Neo (Toyobo, Osaka, Japan) was performed to amplify EGFR (exon 18-21), BRAF (exon 15), KRAS (exon 2), ERBB2 (exon 8, 17, 20), and MET (near exon 14). Two DNA panels (DNA module 1 and DNA module 2), were designed and optimized to detect somatic mutation sensitively and quantitatively with unbiased amplification of these hotspot regions. Forty cycles of 98°C for 10s and 62 °C for 30s were performed to amplify regions on the panel of the DNA module. For RNA assays, first strand cDNA was synthesized by Revertra-ace® (Toyobo), and multiplex PCR using KOD Fx Neo (Toyobo) was performed to detect ALK, ROS1, RET fusion gene variants, and MET exon 14 skipping. Before cDNA synthesis, input RNA was mixed with 9-base random primer (Toyobo) and incubated at 65 °C for 5 minutes for denaturing RNA and for hybridization with primer. The reaction mixture was incubated at 30 °C for 10 minutes, and then at 42 °C for 60 minutes in the cDNA synthesis step. Forty cycles of 98 °C for 15s, 60 °C for 30s, and 68 °C for 10s, followed by extension at 68 °C for 1 minute, were performed to amplify target regions. After purification with AMPure XP (Beckman Coulter Life Sciences), sequence libraries from these PCR products were prepared using the GenNext® NGS Library Prep Kit (Toyobo). All steps were performed according to the manufacturer's protocols. The constructed sequence library was sequenced using MiSeq (Illumina, CA, USA) by paired-end mode (2 x 150 bp). Bioinformatics analysis pipeline is described in the flowchart of Fig. 1. The Illumina adapter sequences were trimmed by Trimmomatic v0.33 and paired-end sequences were joined by FLASH v1.2.11 fastq joining tool. The joined reads were mapped on the target regions of the human genome by BWA aligner v0.7.17 and mutation variant was called by analyzing bam format alignment output by custom programming scripts. Reference sequence for variant call was constructed based on the hg19 human reference genome. HGVD polymorphism database information was used for the discrimination between SNP polymorphism and somatic mutation of variant of unknown significance (VUS).”

For bioinformatics analysis pipeline, a flowchart (Fig. 1) was added in the Methods section.

  1. The authors should provide the respective sequencing depths and QC values of the NGS assays used.

A2) Sequencing depths and QC values were added as supplemental information (Table S1).

  1. Did the cases gene sequence evaluate with the respective ethnicity of the subject?

A3) The following sentence is added in the Methods section:

Reference sequence of amplicon regions for variant call was constructed based on the human reference genome (UCSC hg19). HGVD polymorphism database information was used for the discrimination between SNP polymorphism and somatic mutation of variant of unknown significance (VUS).”

  1. Did the NGS panel follow the ACMG guidance, or other authorities? If yes, which

guidance did the panel follow? Please add information of the guidance followed.

A4) Compliance guidelines were added in the Methods section (page 5), as follows:

“The analytical performance of LC-CP was thoroughly validated according to ICH guidelines (https://www.pmda.go.jp/files/000156867.pdf).”

  1. Please provide information of the ethnicity of the subjects, if available.

A5) We have added the following to the Methods section:

“A prospective observational study of gene panel analysis using cytological specimens is currently underway for Japanese patients with suspected lung malignancies.”

  1. The authors should give details (in each case) the turnaround time of initial diagnosis to result reporting of the NGS test.

A6) The TAT for each case was 18, 10 and 12 days, respectively. However, short TAT was not an aim of this study and therefore, were not described in this study since TATs can be shortened in actual operation. However, if the Reviewer insists, we can add these times to the manuscript.

  1. In case 1, please provide the Oncomine Dx Target Test results by the tissue specimen.

A7) The following was added to the Results section:

“Oncomine™ Dx Target Test confirmed these results (EGFR p.Glu746_Ala750del (COSM ID 6223) with a mutation frequency of 28.7%) using tissue specimen.”

  1. The authors may give the information of the PPV and NPV (positive and negative predictive values) of the assay in this study, and further comments of them.

A8) Sputum specimens were collected in only 3 cases, and PPV and NPV were 100%. Only cases with positive sputum cytopathology were submitted for panel test, and gene mutations were detected for all three cases. Since the number of cases were so few, we do not believe that it is essential to describe PPV and NPV. However, if the Reviewer insists, necessary, we will add this information to the results.

  1. The authors should add the area of limitation, using sputum cytology, and please list the limitation of the current study as a whole.

A9) We added the following limitations to the Discussion.

“For all cases, sputum specimens tended to show lower gene allele frequency when compared to brush specimens when cells were collected directly from the lesion under bronchoscopy. Therefore, it is considered that high sensitivity is an essential requirement for NGS analysis in sputum samples [15].”

“It has been reported that tumor cells account for less than 1% of the cells in sputum [15]. Moreover, since sputum contains mostly normal cells, the technical aspect of a cytopathologist might be affected [16]. Therefore, standardization for the handling of sputum samples may be required in the future.”

  1. The authors may improve the figure resolution of Figure 2.

A10) We enlarged the image and increased the resolution.

  1. In Figure 5, please indicated the time of capture after treatment.

A11) We added when the therapeutic effect was recognized to Figure 5.

  1. The authors may increase the number of relevant references in the article.

A12) We have added the following references:

6) Sing A, Freudenberg N, Kortsik C, et al. Comparison of the sensitivity of sputum and brush cytology in the diagnosis of lung carcinomas. Acta Cytol. 1997 Mar-Apr;41(2):399-408.

7) Warth A. Spread through air spaces (STAS): a comprehensive update. Transl Lung Cancer Res. 2017;6(5):501-507.

8) Toyokawa G, Yamada Y, Tagawa T, et al. Computed tomography features of resected lung adenocarcinomas with spread through air spaces. J Thorac Cardiovasc Surg. 2018;156(4):1670-1676.e4.

15) Thunnissen FB. Sputum examination for early detection of lung cancer. J Clin Pathol. 2003;56(11):805–10.

16) Endo C, Nakashima R, Taguchi A, et al. Inter-rater agreement of sputum cytology for lung cancer screening in Japan. Diagn Cytopathol. 2015;43(7):545–550

  1. As for the case presentation of case 1 to 3, the authors may use a flowchart and timeline to better illustrate the treatments (types, doses and frequency).

A13) We added the following information to Figure 5:

TKI identification, dose, frequency, and when the therapeutic effect was recognized.

  1. The authors should clearly mention the novel findings of the research, in the part of abstract and conclusion

A14) Thank you. We added the following to the Abstract and Conclusion sections.

Abstract

“This is the first report to demonstrate that a novel panel test could detect gene mutations in sputum samples in clinical practice and compare the gene allele ratio with the sample directly collected from the lesion.”

Conclusions

Our new panel test was able to detect gene mutations from sputum samples in clinical practice and we could compare the gene allele ratio with the sample directly collected from the lesion. In the future, a large-scale study testing sputum samples before bronchoscopy will be necessary to confirm diagnosing accuracy.

Reviewer 3 Report

This manuscript by Kei Morikawa et al. describes interesting data on the use of a panel of NGS genes on sputum to enable optimized targeted therapy.

Significant modifications are needed.

Global: 

Prefer passive voice.

Introduction: 

Introduce NSCLC before its first use in the main manuscript.

"Preliminary Results": emphasize this important point by changing the title of the manuscript.

Methods:

"these highly novel findings in advance": this is irrelevant in the methods and irrelevant in the discussion. modify the sentence.

Give a brief description of the analytical process (even if it is described in Ref. 2).

Results: 

The authors should emphasize the advantage of describing three different cases. At this point, the reader could not understand the value of describing this number of cases.

Figures: delete "fig". Provide more detail in the captions, for non-radiologists.

Figure 5 appears before figure 3, please correct.

Discussion: 

italicize "in situ."

The discussion lacks information on the benefit of this technique in the molecular biology landscape. Please complete.

Ethical approval: single consent? or did all patients give consent?

Author Response

This manuscript by Kei Morikawa et al. describes interesting data on the use of a panel of NGS genes on sputum to enable optimized targeted therapy.

Significant modifications are needed.

The Authors would like to thank the Reviewer for raising many details that were missing in our paper. Point-by-point answers are given to all questions and comments below.

Global: 

Prefer passive voice.

A1) I reconfirmed with the native, the manuscript has been written to international journal standards.

Introduction: 

Introduce NSCLC before its first use in the main manuscript.

A2) We added “non-small cell lung cancer” (NSCLC) at first instance.

"Preliminary Results": emphasize this important point by changing the title of the manuscript.

A2) We revised the title to “Preliminary results of NGS gene panel test using NSCLC sputum cytology and therapeutic effect using corresponding molecular-targeted drugs.”

Methods:

"these highly novel findings in advance": this is irrelevant in the methods and irrelevant in the discussion. modify the sentence.

A3) We modified the sentence as follows:

“Among these, sputum was collected for the gene panel prior to bronchoscopy in 3 patients who were found to be positive for sputum cytology.”

Give a brief description of the analytical process (even if it is described in Ref. 2).

A4)

We added a more detailed description of the NGS assay in Methods section, (page 5).

“The experimental process is briefly described below. As a starting material, 5ng of DNA (double stranded DNA) and 5ng of RNA were used for the assay of each module. The LC-CP is composed of two DNA modules and two RNA modules. Therefore, 10ng of dsDNA and 10ng of RNA were set as minimum requirement of starting input material (i.e., the yield of purified nucleotides). For the DNA assay, multiplex PCR using KOD-Plus-Neo (Toyobo, Osaka, Japan) was performed to amplify EGFR (exon 18-21), BRAF (exon 15), KRAS (exon 2), ERBB2 (exon 8, 17, 20), and MET (near exon 14). Two DNA panels (DNA module 1 and DNA module 2), were designed and optimized to detect somatic mutation sensitively and quantitatively with unbiased amplification of these hotspot regions. Forty cycles of 98°C for 10s and 62 °C for 30s were performed to amplify regions on the panel of the DNA module. For RNA assays, first strand cDNA was synthesized by Revertra-ace® (Toyobo), and multiplex PCR using KOD Fx Neo (Toyobo) was performed to detect ALK, ROS1, RET fusion gene variants, and MET exon 14 skipping. Before cDNA synthesis, input RNA was mixed with 9-base random primer (Toyobo) and incubated at 65 °C for 5 minutes for denaturing RNA and for hybridization with primer. The reaction mixture was incubated at 30 °C for 10 minutes, and then at 42 °C for 60 minutes in the cDNA synthesis step. Forty cycles of 98 °C for 15s, 60 °C for 30s, and 68 °C for 10s, followed by extension at 68 °C for 1 minute, were performed to amplify target regions. After purification with AMPure XP (Beckman Coulter Life Sciences), sequence libraries from these PCR products were prepared using the GenNext® NGS Library Prep Kit (Toyobo). All steps were performed according to the manufacturer's protocols. The constructed sequence library was sequenced using MiSeq (Illumina, CA, USA) by paired-end mode (2 x 150 bp). Bioinformatics analysis pipeline is described in the flowchart of Fig. 1. The Illumina adapter sequences were trimmed by Trimmomatic v0.33 and paired-end sequences were joined by FLASH v1.2.11 fastq joining tool. The joined reads were mapped on the target regions of the human genome by BWA aligner v0.7.17 and mutation variant was called by analyzing bam format alignment output by custom programming scripts. Reference sequence for variant call was constructed based on the hg19 human reference genome. HGVD polymorphism database information was used for the discrimination between SNP polymorphism and somatic mutation of variant of unknown significance (VUS).”

For bioinformatics analysis pipeline, a flowchart (Fig. 1) was added in the Methods section.

Results: 

The authors should emphasize the advantage of describing three different cases. At this point, the reader could not understand the value of describing this number of cases.

A5) We have added an explanation to the significance of the three cases in the Discussion.

“Pathologically, spread through air spaces (STAS) is considered to be a positive factor for sputum cytopathology [6]. Clinically predictive factors for STAS include malignant lymphangiopathy, tumor diameter >10 mm, and high standardized uptake value (SUV) for 18F-fluorodeoxyglucose (FDG) - positron emission tomography (PET) image [7]. In this study, we described three cases of different clinicopathological conditions that tested positive for sputum cytopathology. Case 1 showed cancerous lymphangiopathy on endoscopic findings, and case 2 displayed bloodied sputum from the primary lesion without endoscopic lymphangiopathy, while case 3 revealed weak uptake of FDG-PET and CT images suggesting a mucous-producing tumor with glass nodule. There are cases where sputum cytopathology is positive even when it does not correspond to clinical predictors of STAS. Furthermore, we found it significant that it was not only EGFR mutations detected, but also rarer mutations such as MET and KRAS were found in sputum specimens.”

Figures: delete "fig". Provide more detail in the captions, for non-radiologists.

A6) We have revised the figures

Figure 5 appears before figure 3, please correct.

A7) We have corrected the figure order. It seems that the order of the figure like this was mistakenly placed on the system when the draft version was created.

Discussion: 

italicize "in situ."

A8) We italicized “in situ”

The discussion lacks information on the benefit of this technique in the molecular biology landscape. Please complete.

A9) The following has been added to the Abstract and Discussion section.

Abstract

“This is the first report to demonstrate that a novel panel test could detect gene mutations in sputum samples in clinical practice and compare the gene allele ratio with the sample directly collected from the lesion.”

Discussion

First, I would like to explain the significance of explaining the three cases as a case report.

And, at the end of the discussion, I re-emphasized the convenience of sputum testing.

“Currently, mainstream gene panel tests use tissue specimens and blood tests, but if the sputum cytopathology is positive, there are instances when high-sensitivity NGS analysis using sputum specimens is possible, as the least invasive ways for the patients.”

Ethical approval: single consent? or did all patients give consent?

Answer: We obtained consent for all patients. We revised the following sentence.

“Written informed consent was obtained from all patients for publication of this case report and any accompanying images.”

Round 2

Reviewer 3 Report

The manuscript has been revised according to my previous comments.